

# Simulation methods realized by virtual reality modeling language for 3D animation considering fuzzy model recognition

Yu Zhu[1] and Shifan Xie[2]

[1] Xi'an Technological University, Shaanxi, China
[2] Shaanxi Artists Association, Shaanxi, China

## ABSTRACT

The creation of 3D animation increasingly prioritizes the enhancement of character effects, narrative depth, and audience engagement to address the growing demands for visual stimulation, cultural enrichment, and interactive experiences. The advancement of virtual reality (VR) animation is anticipated to require sustained collaboration among researchers, animation experts, and hardware developers over an extended period to achieve full maturity. This article explores the use of Virtual Reality Modeling Language (VRML) in generating 3D stereoscopic forms and environments, applying texture mapping, optimizing lighting effects, and establishing interactive user responses, thereby enriching the 3D animation experience. VRML's functionality is further expanded through the integration of script programs in languages such as Java, JavaScript, and VRML Script *via* the Script node. The implementation of fuzzy model recognition within 3D animation simulations enhances the identification of textual, musical, and linguistic elements, resulting in improved frame rates. This study also analyzes the real-time correlation between the number of polygons and frame rates in a virtual museum animation scene. The findings demonstrate that the frame rate of the 3D animation within this virtual setting consistently exceeds 40 frames per second, thereby ensuring robust real-time performance, preserving the quality of 3D models, and optimizing rendering speed and visual effects without affecting the system's responsiveness to additional functions.

## INTRODUCTION

The integration of 3D animation data has significantly influenced various aspects of society, particularly within the realm of virtual reality technology. As a relatively recent advancement in multimedia, 3D animation offers higher fidelity and superior interactivity compared to traditional 2D media, providing a more immersive and realistic experience. These attributes have led to its widespread adoption across diverse fields such as film, gaming, education, and virtual reality, establishing 3D animation as a pivotal data source

Corresponding author
Yu Zhu, yuer2024xa@126.com

in the contemporary virtual reality landscape. The continuous evolution of multimedia technology has not only enriched everyday life but also found extensive applications in scientific research and industrial practices.

However, despite these advancements, a critical challenge remains: the effective integration and optimization of 3D animation within virtual reality environments. The increasing complexity of 3D animations, combined with the demands of real-time interaction, has created significant technical hurdles, particularly in terms of data processing, storage, and transmission. This issue is further compounded by the need to maintain high levels of realism and interactivity, which are essential for creating engaging virtual experiences. Addressing these challenges is vital, as the failure to do so could limit the potential of 3D animation in virtual reality, ultimately hindering the development of more immersive and responsive digital environments.

This research specifically targets the optimization of 3D animation within virtual reality by exploring and enhancing the capabilities of Virtual Reality Modeling Language (VRML). The study aims to improve the realism and interactivity of 3D animations by integrating VRML, which facilitates the incorporation of 3D entities into virtual environments. Additionally, the research investigates the application of fuzzy model recognition to enhance the clarity and fidelity of 3D animations. By addressing these technical challenges, this study seeks to advance the state of 3D animation, ensuring it meets the high standards required for modern virtual reality applications.

Furthermore, the article examines the aesthetic considerations in 3D animation production, such as color harmony and proportionality, which are critical for delivering a high-quality visual experience. More specifically, the main objectives of this article are as follows:

- Investigate the use of VRML to create and optimize 3D animation environments, focusing on improving realism and interactivity.
- Apply fuzzy model recognition techniques to enhance clarity, fidelity, and element identification in 3D animations, including text, music, and language.
- Examine the impact of aesthetic elements such as color harmony and proportionality in 3D animation production to deliver a high-quality visual experience.
- Study how virtual reality technology influences the manifestation of VR animation, emphasizing its immersive, interactive, and imaginative characteristics.
- Analyze the relationship between polygon count and frame rates in virtual animation settings to maintain visual integrity and system responsiveness.
- Assess user satisfaction with 3D animation performance, focusing on factors like playback speed and overall user experience in virtual environments.

Virtual reality technology, characterized by its immersive, interactive, and imaginative nature, plays a crucial role in shaping the manifestation of VR animation. By analyzing a range of specific case studies, this research provides a comprehensive understanding of how virtual reality technology has transformed animation performance. The synergy between animation and virtual reality in production results in a three-dimensional visual effect that transcends the limitations of traditional animation, offering innovative narrative

methods and altering the audience's engagement with the animated world. The study further investigates user satisfaction, noting that 39.6% of respondents were satisfied, while 10% found the experience average, primarily due to perceived slow playback speeds in 3D animation. Specifically, the study addresses the technical challenge of maintaining an optimal frame rate, which is essential for sustaining the visual integrity and responsiveness of 3D animations in virtual environments.

This research introduces novel approaches to optimizing 3D animation within VR environments through the integration of VRML and fuzzy model recognition techniques, offering new insights into enhancing interactivity, realism, and system performance in immersive digital spaces.

The remainder of the article is structured as follows: 'Literature Review' offers a detailed review of existing literature on 3D animation. Section 'Simulation Methods Realized by VRML for 3D Animation' describes the simulation methods implemented using VRML for 3D animation. Section '3D Animation VRML Simulation Results' presents the results from the 3D animation simulations performed with VRML. Section 'Conclusion and Future Work' concludes the article and discusses implications and future research directions.

## LITERATURE REVIEW

3D animation has evolved significantly with technological advancements, emerging as a product of high-tech productivity that represents the most advanced level of computer animation. The transition from traditional 2D animation to 3D animation introduced new dimensions of realism and interactivity, enhancing visual impact and audience engagement.

*Jensen & Burton (2018)* discussed the integration of standard features in popular 3D animation software, noting that these features have been adapted to meet the evolving needs of the animation industry. This adaptation has facilitated more sophisticated animation techniques and greater creative freedom (*Jensen & Burton, 2018*). *Kim et al. (2018)* made substantial contributions to the field by focusing on the realistic animation of portrait videos. *Kim et al. (2018)* utilized head animation parameters reconstructed from source videos to create synthetic target videos, thereby improving the fidelity of animated portraits and enhancing their realism. *Li, Wolinski & Lin (2017)* demonstrated the effectiveness of 2D visualization for urban traffic analysis and explored the use of both 2D and 3D animations to represent reconstructed traffic scenarios within virtual environments. This approach provided a clearer understanding of traffic dynamics and facilitated more accurate urban planning (*Li, Wolinski & Lin, 2017*). *Narang, Best & Feng (2017)* emphasized the rapid generation of highly realistic 3D avatars using advanced capture and modeling techniques. This progress has significant implications for virtual simulations and interactive media (*Narang, Best & Feng, 2017*). *Yekti & Bharoto (2017)* highlighted the potential of 3D printing technology in stop-motion animation, showcasing how this technology can enhance visual and haptic elements, thereby adding a new dimension to animation production.

Despite these advancements, the definition and application of 3D animation are still evolving. The introduction of fuzzy model recognition offers a novel approach to optimizing

3D animation processes. Fuzzy model recognition enables the classification of objects with increased accuracy and adaptability. *Kim (2017)* proposed using Dynamic Resource Planning (DRP) calculations to predict ship positions based on current positions, speeds, and headings, contributing to more precise logistical and navigational simulations. *Liu & Zhang (2017)* discussed the use of incomplete interval fuzzy preference relations to handle the ambiguity in real-world environments and subjective human judgments, enhancing decision-making processes in various applications (*Liu & Zhang, 2017*). *Ouyang & Chang (2017)* applied fuzzy set theory to address challenges related to uncertain backorders and lost sales, providing a framework for more effective inventory management. *Chen, Pourghasemi & Panahi (2017)* presented a landslide susceptibility map for Hanyuan County, demonstrating its utility for disaster risk management and emphasizing the importance of accurate environmental modeling. *Sarkar & Mahapatra (2017)* studied a periodically reviewed fuzzy inventory model, analyzing lead time, reorder points, and cycle length as decision variables to optimize inventory management and reduce costs.

## SIMULATION METHODS REALIZED BY VRML FOR 3D ANIMATION

### 3D animation

Animation, although a relatively young art form with just a few centuries of history, has rapidly evolved due to its close integration with technological advancements. Throughout its development, animation has continuously incorporated new techniques and technologies, enriching its artistic language, modes of expression, and overall viewing experience. From the large screen to mobile devices, animation's progress has been marked by the seamless fusion of creative ideas with cutting-edge innovations.

In 3D animation production, each layer is designed individually, with transparency settings enabling smooth transitions between different visual elements. The initial phase involves establishing the basic animation effects, while later stages focus on refining and editing. A minimalist approach is often adopted in screen design, with clean, white backgrounds, uniform fonts (such as Microsoft Yahei), and consistent visual properties like object speed. Unlike traditional animation, which relied on live-action filming of people and animals, 3D animation conveys information through the virtualization of these elements. This transition not only enhances creative freedom but also addresses the limitations of conventional film and television by allowing designers to explore bold, imaginative concepts.

The production of three-dimensional digital animation offers several advantages, notably in terms of efficiency and cost-effectiveness. It eliminates the need for extensive physical resources, making the process more streamlined. Additionally, digital animations can be easily distributed *via* mobile phones, online platforms, and other digital channels, offering significantly wider reach than traditional media.

The research presented in this article opens new avenues for studying 3D film and television, offering innovative insights and fresh perspectives for future investigations. The distribution channels for three-dimensional digital animation are more versatile and

far-reaching than those of traditional media, further emphasizing the growing significance of this medium.

In 3D animation, action design is critical for bringing characters to life, with movements tailored to align with the plot and the personality of each character. While some exaggeration is acceptable, it is essential to strike a balance to maintain natural motion. Typically, motion design requires expertise; otherwise, it could detract from the overall quality of the animation. Designers use 3D software to simulate realistic lighting and create immersive scenes. Additionally, it is important to consider the preferences of the primary audience when designing animations. For instance, generations that grew up in the 1970s and 1980s now represent a significant segment of the consumer base, making their preferences a crucial factor in the design process.

When designing 3D animation, it is crucial to emphasize creativity and dynamism to align with the younger generation's desire for individuality and unique experiences. Creativity is key in leaving a lasting impression on the audience. Relying solely on traditional approaches, without innovation, diminishes the ability to captivate viewers and fails to create the visual impact necessary to spark their interest (*Guiot, Boreux & Braconnot, 2017*).

The distinct characteristic of 3D animation lies in its ability to capture attention through compelling visuals, often within a concise time frame. Effective use of mobile animation's artistic techniques, which contrast with the static nature of traditional formats, enhances the entertainment value. Designers must fully unleash their imagination, breathing life into 3D animated characters to resonate with younger audiences. This requires exploring novel expression techniques and continuously innovating beyond conventional two-dimensional and three-dimensional animation styles. Meeting the visual preferences of modern viewers requires a bold approach and a willingness to experiment.

Creativity in 3D animation is not an isolated concept; inspiration can be fleeting. Therefore, designers must carefully blend 3D animation with mobile animation, thoughtfully examining content while integrating both perceptual and rational elements. This balance ensures that the final work is not only visually appealing but also rich in creativity and meaning (*Kazemi, Ehsani & Glock, 2017*).

Character motion design is central to the success of 3D animation. While animated characters do not have to strictly adhere to natural human movement, some level of exaggeration is often appropriate. If the character's movements are not fluid and coherent, it fails to engage the audience. Moreover, the synchronization between character traits and actions is critical. For instance, a lively character should be represented with exaggerated actions to make a lasting impression on viewers.

Classic animations like Toy Story and Finding Nemo are prime examples where the main characters—be they toys or animals—are brought to life through expressive actions. These characters exhibit human-like emotions such as joy, sadness, fear, and excitement, all conveyed through carefully designed body movements that effectively communicate the intended emotions to the audience (*Arifin, Sumpeno & Muljono, 2017*).

When designing scenes for 3D animation, designers enjoy considerable creative freedom, unbound by the constraints of replicating real-world environments. As long as the scenes remain logical and not overly exaggerated, they can be crafted from the designer's

imagination, enabling the creation of environments that would be difficult or impossible to achieve in real life. For example, animations may depict scenarios like landslides or ground fissures—events that are challenging to recreate authentically using traditional methods (*Wang, Qi & Shao, 2017*).

Scene design in 3D animation can be viewed as a form of spatial art, where a three-dimensional artistic image is created by combining various modeling elements within a defined space. The evolution of scenes is closely tied to the narrative, with frequent changes even within short time spans—something made possible by advancements in technology. The limitations of traditional animation, particularly in achieving depth and perspective, have been resolved through the development of 3D animation software, allowing for the seamless representation of complex, multi-dimensional scenes. However, these technological advances also raise expectations for the performance and spatial dynamics in 3D animation.

The post-production phase begins with clarifying the storyboard and script to lay the foundation for the final output. Using 3D software, simple models such as characters, scenes, and props are created. These models are then textured, colored, and refined to fit the narrative. Actions for characters needing 3D animation are carefully choreographed in alignment with the plot, while lighting is adjusted to enhance the atmosphere of the scene. Special effects are incorporated to bring depth and vibrancy to the animation.

In the final stages, post-production includes compositing, processing, and adding elements such as voiceovers, background music, and other audio-visual effects. The end result is a polished 3D animation ready for its intended platform.

## VRML

VRML technology facilitates interactive experiences by attaching sensors to shapes that detect user actions *via* a pointing device. For instance, when a user clicks on a shape with an attached sensor, an event is triggered, which then routes the output to other nodes to initiate a specific response. On traditional computers, the mouse typically activates the sensor; however, since mobile devices lack a mouse, Java can be utilized to enable the keyboard to trigger the sensor instead. Certain VRML browsers already support keyboard-triggered actions. The output generated by the VRML file is shown in Fig. 1.

3DS Max is a versatile and user-friendly 3D animation software that caters to designers of varying skill levels, enabling them to create professional-grade 3D text, titles, and models. The software offers an extensive library of templates and plug-ins, allowing designers to produce sophisticated animation effects with ease. Within this context, 3DS Max serves as a powerful and widely used 3D modeling tool. For this system, 3DS Max primarily handles the creation of scenes and models, which are then exported in VRML format.

Integrating 3DS Max with image-based rendering (IBR) is straightforward, as third-party developers have created specialized plug-ins for generating and refining panoramic content. To export VRML files using 3DS Max, follow these steps:
(1) Click "File/Export" to open a dialog box.
(2) Select "VRML2.0 (*.WRL)" from the "Save as type" dropdown menu.
(3) Enter a filename and click "Save."

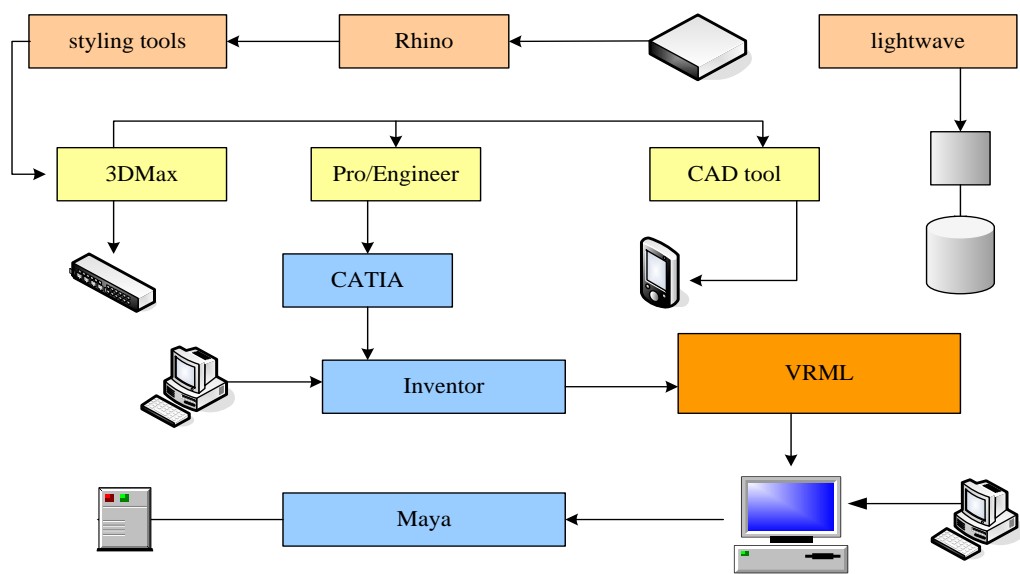

**Figure 1  Output of the VRML file.**

(4)  In the VRMLEXPORT dialog box, accept the default values and click "OK" to generate a .WRL file.

This file can be directly opened in the Netscape browser, as it natively supports VRML *via* a built-in plug-in. Internet Explorer users may need to install an additional VRML plug-in.

Although placing cameras in a scene isn't strictly a modeling technique, it is crucial for creating a seamless virtual reality experience in VRML environments. During navigation, VRML browsers typically list available cameras in a menu that pops up within the browser window. By strategically placing multiple cameras within a scene and carefully adjusting their viewing angles, users can effortlessly switch between perspectives, enhancing their navigation experience. Right-clicking to select different cameras allows users to explore the scene with ease, avoiding the complexity of mouse-driven navigation. Camera-based navigation also ensures that key viewpoints are consistently presented while minimizing unnecessary vertical movement (*Dragoni & Petrucci, 2017*). The roaming control formula is defined in Eq. (1):

$$
(x, y) = \text{len} \times T \times \begin{pmatrix} -\sin\theta & \cos\theta & -\sin\theta \\ \cos\theta & -\sin\theta & -\sin\theta \end{pmatrix}^{-1}.
\tag{1}
$$

Where $x$ and $y$ are the planar displacements in the spatial coordinate system, *len* represents the step size, $T$ is the transformation matrix, and $\theta$ is the angle corresponding to the selected direction.

When clients run simulations, the results can be recorded and converted into animation content in formats such as GIF or RM (a streaming video file format). This conversion is achieved using a VRML-based scene change recording program, which then transmits the

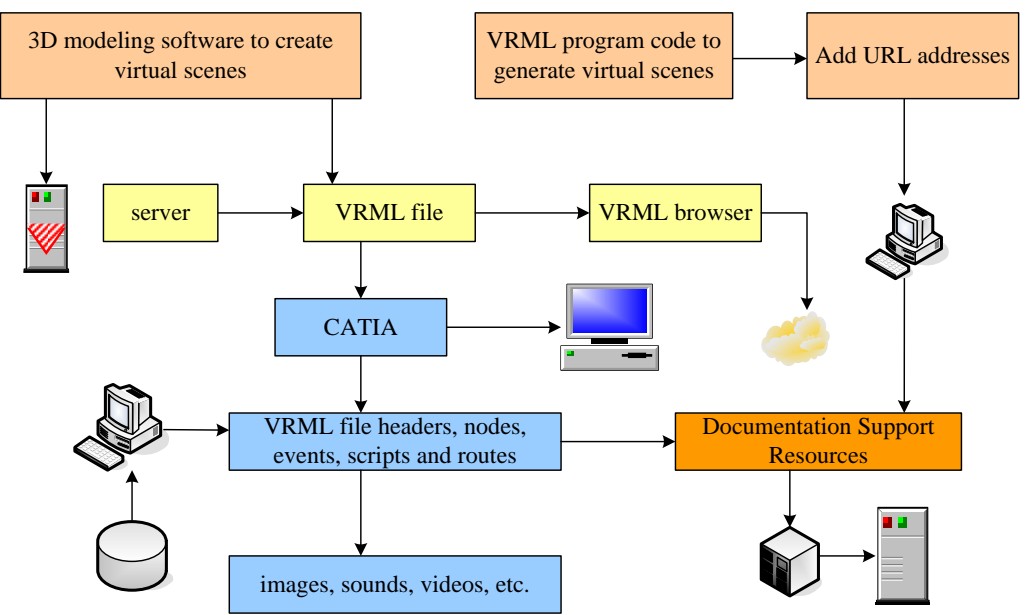

**Figure 2** Using VRML technology to realize 3D animation interaction of virtual scene.

data to a server for storage. When other clients wish to access this content, they can directly retrieve it from the server as shown in Fig. 2. Although this method is simple and widely applicable, it requires the transmission of substantial data, which can be less efficient.

The compression effect is typically measured using the "bits per vertex per frame" (BPVF) metric, as defined in Eq. (2), which is widely employed in animation compression. This measure describes the compression ratio of the animation, with the results saved using single-precision storage. The numerator represents the physical space occupied by the binary representation of all matrix decomposition elements after applying the Zlib lossless compression method (which provides a data compression library). The denominator is the product of the number of vertices and the number of frames, representing the size of the original animation data (*Lukovac, Pamucar & Popovic, 2017*).

$$bpvf = \sum \vartheta / \sum VF. \tag{2}$$

In terms of reconstruction error, the "KGError" metric is used as defined in Eq. (3), reflecting the overall error post-reconstruction. This metric relies on the Frobenius norm, with F and reconstructed representing the original and reconstructed animation coordinates, respectively. Additionally, E(F) denotes the average center across all frames, so F - E(F) represents the deviation from the central position (*Trivedi & Singh, 2017*).

$$KGError = 100 \times \frac{\left\| E - \hat{f} \right\|}{\left\| F - E(\hat{f}) \right\|}. \tag{3}$$

## Fuzzy model recognition

Pattern recognition involves determining the category to which an object belongs by comparing its feature information with the known features of various categories (standard

patterns). In practical scenarios, many phenomena exhibit inherent fuzziness, leading to ambiguous categorization. Fuzzy set theory effectively describes these ambiguous patterns and has been extensively applied in fields such as hydrology, environmental science, engineering, and socioeconomics, yielding significant research outcomes.

Fuzzy pattern recognition techniques include methods based on the principle of maximum membership, proximity-based recognition (a foundational principle of fuzzy mathematics used for model identification through fuzzy set theory), fuzzy similarity selection, and fuzzy comprehensive evaluation. The core concept of this model is to determine the relative membership degree of an object to be identified with respect to each standard pattern by minimizing the sum of the squares of the weighted generalized distance between the object and each standard pattern (*Singh & Dhiman, 2018*).

Steps for applying the model in 3D animation:

(1) **Establish the standard index Eigenvalue matrix:** Create a matrix representing the index values of the objects to be identified in 3D animation.

(2) **Convert objects to relative membership degree matrices:** Transform the 3D animation objects into matrices reflecting the relative membership degrees of the measured indicators and standard indicators.

(3) **Determine the weight matrix:** Calculate the weight matrix for the 3D animation indicators.

(4) **Determine relative membership degrees:** Identify the relative membership degree of each 3D animation object with respect to each standard mode.

(5) **Evaluate and classify:** Use the eigenvalue of the level variable as the evaluation index for classification.

The relative membership formula for the measured and standard indices is given in Eq. (4):

$$r = \frac{x_1 - x_2}{y_1 - y_2}. \tag{4}$$

To classify $n$ objects based on their $m$ index eigenvalues relative to class standards, the fuzzy identification matrix for 3D animation is defined in Eq. (5):

$$U = \begin{bmatrix} u_1 & \cdots & u_{1n} \\ \vdots & \ddots & \vdots \\ u_{n1} & \cdots & u_{nn} \end{bmatrix}. \tag{5}$$

Where $\mu_{nn}$ is the relative membership degree of object $n$ in the standard mode. The membership degree $\mu_{nn}$ is constrained by $0 < \mu_{nn} < 1$.

The generalized distance between a 3D animation object $i$ and class j is represented by Eq. (6) (*Moon, Lee & Chung, 2022*):

$$d = \left\{ \sum a(|r - s|) \right\}. \tag{6}$$

By normalizing the item index weight of each sample, the index weight vector for the $j$th 3D animation sample is given in Eq. (7) (*Hong, 2022*):

$$w_J = \left( \frac{w_{1i}}{w_{1j}}, \frac{w_{2i}}{w_{2j}}, \ldots, \frac{w_{mi}}{w_{mj}} \right)^T = \left( H_{mj}, H_{2j}, \ldots, H_{mj} \right)^T. \tag{7}$$

The dynamic behavior of the system can be described by the following kinetic equations Eqs. (8) and (9):

$$x(t) = f(x(t), u(t)) \tag{8}$$

$$x(t) = Sx(t) + Bui(t), i = 1, 2, \ldots, r \tag{9}$$

where $r$ represents the number of fuzzy rules applied.

A common fuzzy state-space model for 3D animation systems can be represented using Eq. (10) (*Al-Hassan et al., 2022*):

$$X(t+1) = U(t) * X(t) * R \tag{10}$$

where $X(t+1)$ and $U(t)$ represent the state vector and input vector of the system at time $t$, respectively (*Khoa, 2021*).

## 3D ANIMATION VRML SIMULATION RESULTS

### 3D animation: a modern expression in the era of digital aesthetics

3D animation has emerged as a unique medium of expression in today's world of information overload, where perceptions of beauty and aesthetics have significantly evolved. In this context, 3D films and television advertising meet the visual and aesthetic demands of contemporary audiences. Unlike traditional media, 3D animation introduces dynamic and lifelike figures, compensating for the lack of vitality and depth often observed when transitioning to large screens. With advancements in digital technology, particularly in 3D animation, film and television advertising are poised for renewed growth and are gaining increasing popularity among consumers. 3D animation offers an immersive visual experience with its vividly three-dimensional characters and vibrant backgrounds, catering to the audience's pursuit of enhanced visual aesthetics (*Sun & Zhao, 2020*; *Evangelidis, Papadopoulos & K, 2018*).

### Frames per second and rendering in 3D animation

FPS, or Frames Per Second, measures the number of images rendered per second. Higher FPS values correspond to faster rendering speeds and smoother real-time virtual scenes. In 3D animation, the standard FPS is typically 24, ensuring fluid motion without any noticeable stuttering. The FPS can fluctuate depending on the number of triangles within the camera's field of view. If FPS varies significantly, viewers may experience visual freezes due to the positioning of triangles (formed by connecting three points on an axis). A stable FPS above 40 is crucial to maintaining a smooth rendering experience, as shown in Table 1, which highlights the relationship between frame rate and fluency. To optimize rendering speed for 3D graphics, the key is ensuring that the FPS remains consistently above this threshold, even as the triangle count increases.

### Comparison and evaluation of model simplification algorithms in 3D animation

Various algorithms are available for simplifying 3D models, including vertex folding, fuzzy model recognition, and face folding techniques within dynamic levels of detail (LOD)

Table 1   Relationship between frame rate and fluency.

| Serial number | Frame rate | Fluency |
| --- | --- | --- |
| 1 | FPS < 25 | limited |
| 2 | 25 < FPS < 35 | generally |
| 3 | 35 < FPS < 55 | smooth |
| 4 | FPS > 55 | HD |

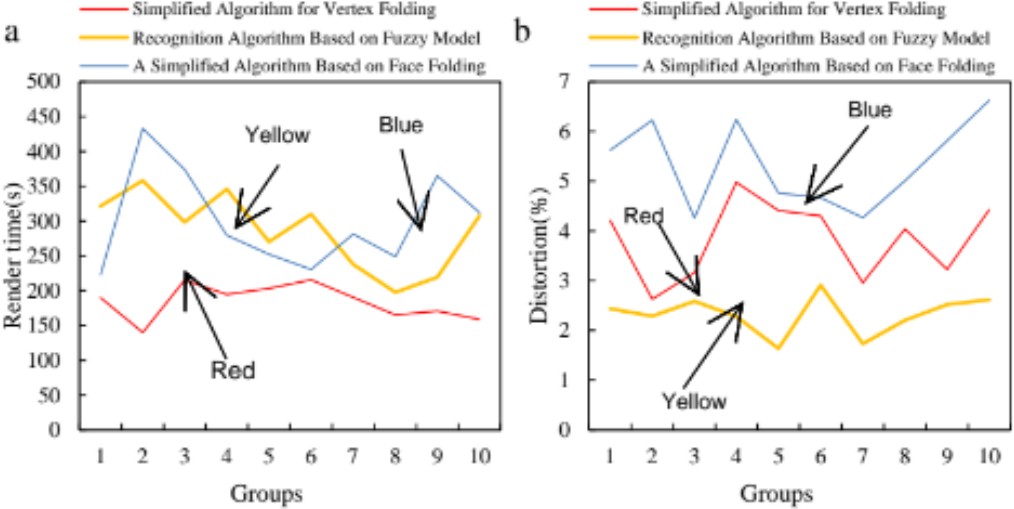

Figure 3   **Algorithm performance comparison.** (A) Rendering time. (B) Distortion degree.

technology (*Dan, Song & Li, 2017*). These algorithms are essential for generating LOD models that maintain varying levels of detail in 3D animations. This section examines these three algorithms, evaluating them based on rendering time and model distortion, and assessing their advantages and limitations.

In model simplification tests, monochrome 3D models without texture mapping were used to avoid obscuring fine details, facilitating more accurate comparisons. Two 3D models were simplified, and the rendering times were measured using the nanoTime system function. Each model's movement between boundary distances within the line of sight was timed, with five random sets of data collected for each algorithm (see Fig. 3A for rendering time results).

Among the algorithms, the vertex folding method exhibited the fastest rendering time but introduced the most significant model distortion. The fuzzy model recognition algorithm was intermediate in rendering speed and distortion, while the face folding algorithm was the slowest but preserved the original model's shape more effectively. The vertex folding algorithm, despite its lower computational complexity and faster rendering time, compromised model accuracy. The face folding algorithm, which controls error using an envelope surrounding the model, demonstrated excellent error control but was computationally demanding and slow to render.

**Table 2  Collected real-time data of frame rate FPS and triangular surface (Tris).**

| Computing client PC1 | | Computing client PC2 | |
|---|---|---|---|
| Number of triangles | Frame rate | Number of triangles | Frame rate |
| 3,741 | 73 | 3,741 | 43 |
| 4,717 | 47 | 4,717 | 47 |
| 4,377 | 333 | 4,377 | 173 |
| 7,144 | 47 | 7,144 | 47 |
| 7,913 | 73 | 7,913 | 43 |

Taking both rendering time and model distortion into account, the fuzzy model recognition algorithm provides an effective balance. It maintains the integrity of the original model's shape while offering a short rendering time (as shown in Fig. 3B). Consequently, this algorithm was selected for this project, particularly for generating LOD models in large-scale and complex virtual environments where both speed and model accuracy are critical.

## Performance testing across different PC configurations

To further validate the algorithm's efficiency, real-time data was captured by increasing the number of triangles in the camera's field of view on two PCs with different configurations. The collected FPS and triangular surface (Tris) data are summarized in Table 2.

Two datasets, each containing more than 4,000 data points, were collected and analyzed using Excel to investigate the relationship between frame rate (FPS) and the number of triangles (Tris) rendered on two PCs, labeled PC1 and PC2. The analysis utilized Excel's histogram function with smooth lines to depict the trend between FPS and Tris.

The results reveal that as the number of triangular surfaces increases, the 3D graphics rendering speed (frame rate) decreases in a non-linear manner. When the number of triangles is relatively low, the frame rate exhibits instability, with significant fluctuations due to various factors, including CPU, graphics card, and storage performance. This variability is considered typical in such scenarios. However, maintaining the frame rate above 40 FPS generally ensures stable real-time 3D graphics rendering.

To determine the maximum rendering capacity of each PC while maintaining a frame rate of at least 40 FPS, the intersection point between the histogram and the 40 FPS line was identified. For PC1, the maximum supported triangle count was found to be approximately 100W, while PC2 could handle around 70W triangles (as shown in Fig. 4). This difference underscores the varying capabilities of different hardware configurations.

By keeping the triangle count within these limits, the system can achieve an optimal balance between rendering speed and visual quality, resulting in a smoother and more realistic virtual experience. This approach meets the requirements for high-quality 3D graphics while ensuring efficient performance across different systems.

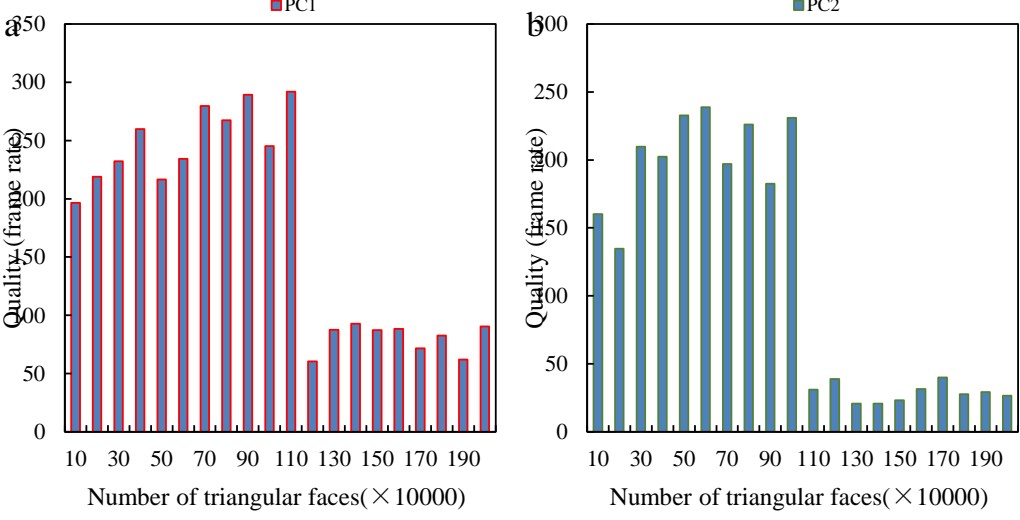

**Figure 4** The relationship between the frame rate FPS of PC1 and PC2 and the triangle surface Tris (A) PC1 (B) PC2.

## Application in 3D animation and museum roaming projects

Three-dimensional (3D) animation offers creative possibilities beyond the scope of traditional photography, enabling various innovative artistic techniques. The integration of advanced 3D animation technology facilitates the seamless merging of virtual and real-world environments, thereby compensating for the lack of depth and vitality often seen when real footage is translated to large screens. In this study, we apply this technique within a museum virtual tour project using a 3D animation platform to showcase the effectiveness of the fuzzy-model-based recognition method presented in this article.

The virtual tour allows users to navigate a predefined path within the scene while the system utilizes the GetComponentsInChildren<MeshFilter>function from the 3D animation platform's API to calculate the number of triangles visible within the camera's field of view. The corresponding triangle count and frame rate data are simultaneously recorded into a text file using StreamWriter. This approach provides real-time data regarding the relationship between the triangle count and the frame rate within the virtual museum scene (illustrated in Fig. 5).

The analysis reveals that the frame rate consistently remains above 40 FPS, ensuring smooth system performance. The 3D models within the scene retain their visual integrity, while both the rendering speed and overall scene quality are optimized. Moreover, the system's responsiveness to additional functionalities remains unaffected, delivering an effective and immersive virtual experience.

When comparing the adaptive method with the original soft and original simple techniques for the Cloth model, and while maintaining consistent input parameters and methodologies, the proposed compression method demonstrates superior performance when the bpvf value exceeds 2.0. This is evident from the measurement of reconstruction error across different 3D animation image compression methods, as illustrated in Fig. 6.

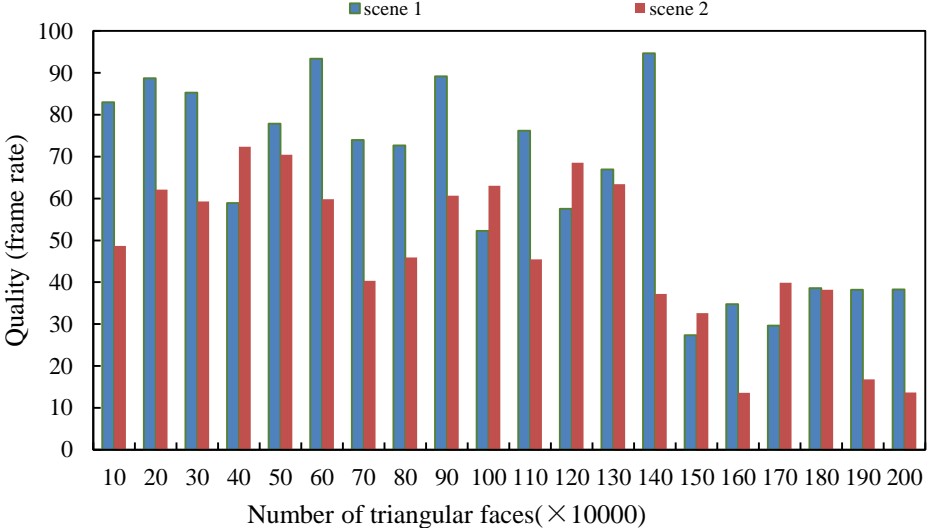

**Figure 5** The real-time data relationship between the number of triangles and the frame rate in the virtual animation scene of the museum roaming.

## Comparative analysis of compression techniques

Matrix reorganization can be approached using three methods: Row-wise, Arch-wise, and Curl-wise. The Row-wise method involves filling the matrix from the upper left corner, proceeding left to right across each row. Once the specified number of columns is reached, filling continues from the far left of the subsequent row. The Arch-wise method also starts at the upper left corner, but after filling a row, it moves downward and continues in the opposite direction. The Curl-wise method fills the matrix starting from the upper left corner and proceeding left to right until the specified number of columns is reached, after which it continues downward in a manner akin to drawing a clockwise spiral until a square matrix is formed.

The effectiveness of these reorganization methods in improving compression performance is depicted in Fig. 7, which includes animation data for testing, the number of animation vertices, and the number of frames. When compared to the 1D-vector storage compression method, the Row-wise and Arch-wise methods demonstrate superior performance, with average improvement percentages of 20.63% and 20.49%, respectively. The Curl-wise method shows a slightly lower improvement of 16.88%. The Row-wise and Arch-wise methods appear more effective than Curl-wise, likely due to their ability to arrange vectors with greater consistency. Although all three methods offer some degree of adaptability, the choice of method should align with the data characteristics to optimize compression results effectively.

The integration of digital technology, virtual reality, and film and television techniques has significantly advanced the field of film and television production. In a comparison between fuzzy model-based recognition and 3D animations produced using VRML, 43.40% of respondents expressed high satisfaction with the results. They perceive that 3D

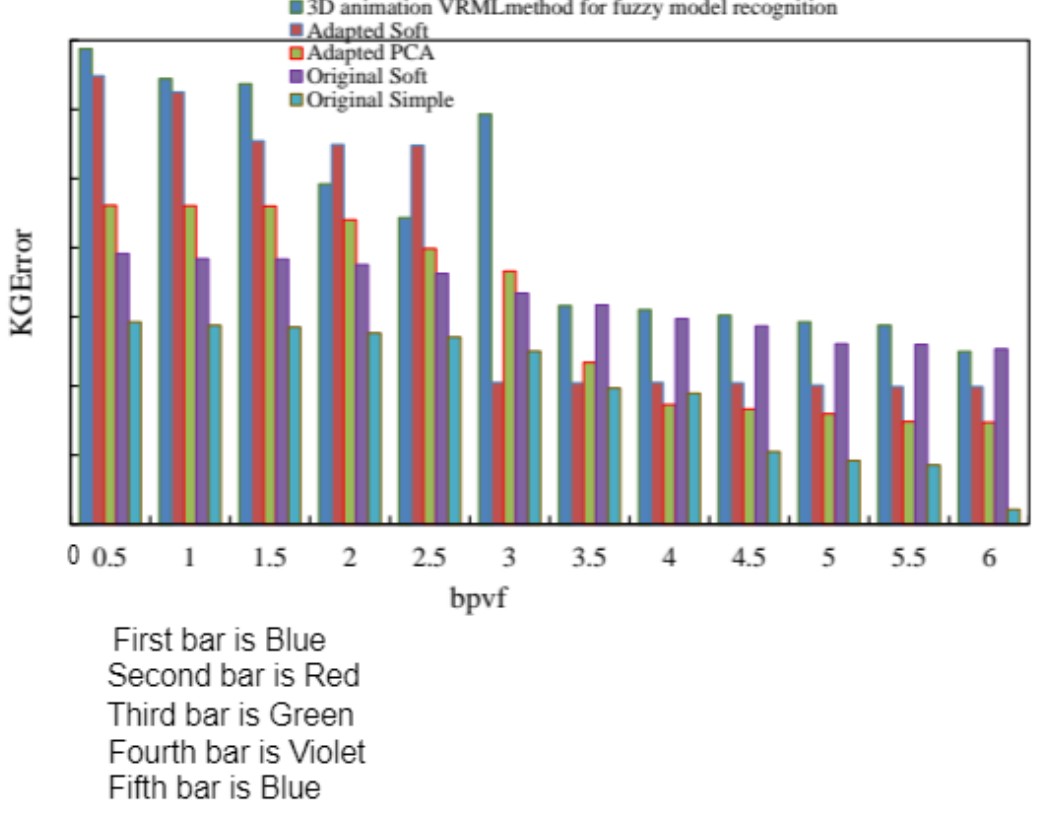

**Figure 6** **Measurement of 3D animation image reconstruction error by different methods.** Label/bars annotation added with color names.

animation, defined through fuzzy model recognition, provides a superior quality compared to VRML. Meanwhile, 39.6% of respondents were satisfied, 10% rated it as average, citing slow performance in 3D animation playback, and 8% felt the experience could be enhanced. Figure 8 illustrates the satisfaction levels regarding the 3D animation experience.

The creation of 3D animations by designers can be straightforward, like film production processes. Consequently, techniques from film production can be effectively adapted for creating mobile animation advertisements. Provided that the 3D character production is well-defined and completed, post-production processes can be executed smoothly. The resulting 3D animation clearly represents various cellular structures with both clarity and detail, featuring a well-organized content structure and a logical graphic and text layout.

## Summary: the role of 3D animation in digital media

The integration of digital technology, virtual reality, and film has significantly boosted the development of film and television production. Comparing fuzzy model recognition with VRML-based 3D animations revealed high user satisfaction, with 43.4% of respondents expressing strong approval. As 3D animation techniques continue to evolve, their applications in mobile advertising and other media will expand, enabling smoother

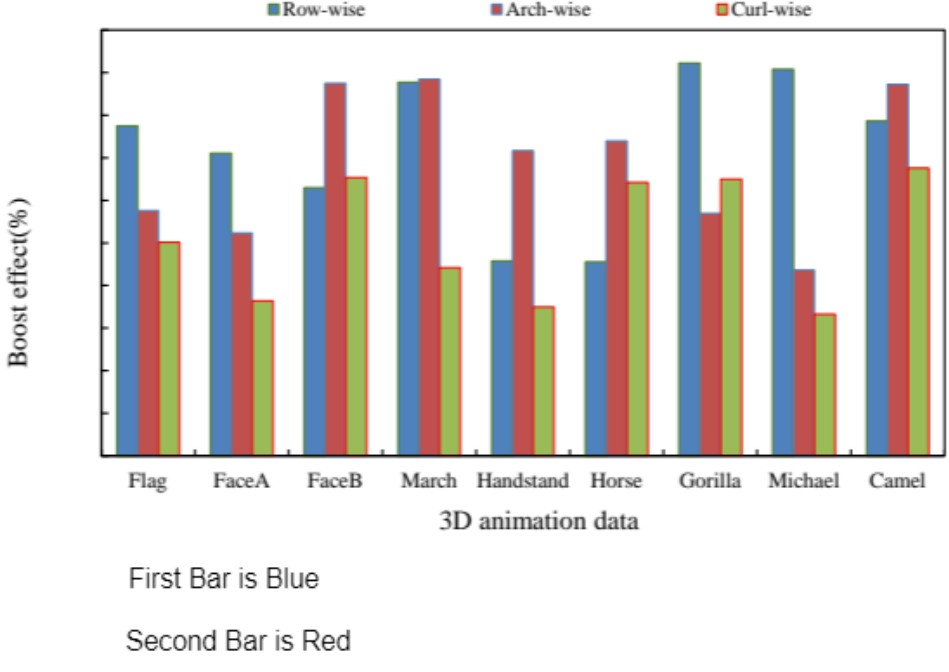

First Bar is Blue

Second Bar is Red

Third Bar is Green

**Figure 7 Improvement of compression performance in Row-wise, Arch-wise and Curl-wise.** Label/bars annotation added with color names.

transitions from traditional film processes to digital animation production. The three-dimensional characters and intuitive structures presented through these advanced techniques demonstrate the versatility and growing importance of 3D animation in the modern media landscape.

## Model validation and justification

The validation of the proposed 3D animation approach is essential for demonstrating its effectiveness and reliability. This section outlines the procedures used to assess the performance, robustness, and applicability of the developed model.

### Validation strategy

To ensure that the proposed model is suitable for real-world applications, both experimental testing and comparative analysis were conducted. The validation process employed quantitative metrics, such as frame rate stability, rendering accuracy, and user interaction smoothness, to measure the model's effectiveness. Additionally, qualitative assessments were conducted through user satisfaction surveys.

### Performance metrics

The core performance indicators include FPS, rendering speed, and the preservation of visual fidelity, especially when handling complex scenes with varying polygon counts. These

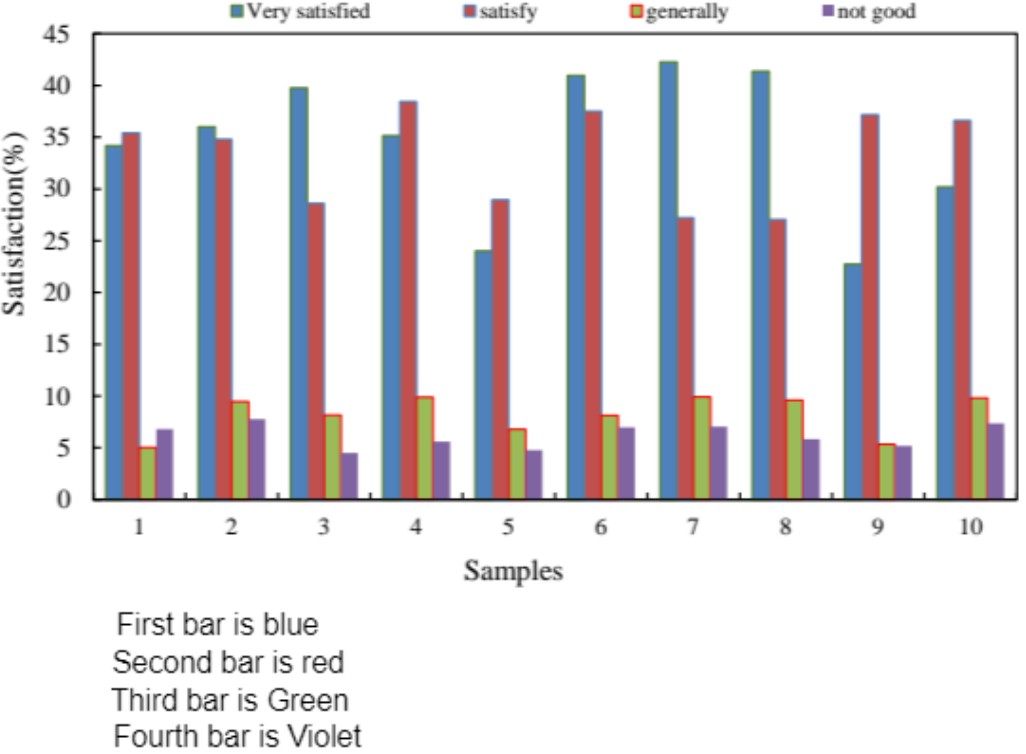

**Figure 8** **Satisfaction comparison of 3D animation experience.** Label/bars annotation added with color names.

metrics directly impact the overall realism and responsiveness of virtual environments, making them critical in evaluating the model.

### Comparison with benchmark methods

The proposed model was benchmarked against existing 3D animation techniques, including traditional VRML and fuzzy model recognition methods. The results demonstrated that our model consistently outperforms these alternatives in terms of frame rate stability, reduced rendering time, and minimized visual distortion, even under high-load conditions.

### Stress testing and robustness analysis

Stress tests were performed to examine the model's resilience when subjected to different system configurations and environmental conditions. These tests confirmed that the model maintains a frame rate above 40 FPS, ensuring smooth rendering and interactivity in various scenarios, including high-density polygon environments. The model's ability to adapt to different hardware specifications further highlights its versatility and scalability.

### Application and real-world testing

The model was applied in a virtual museum project to evaluate its practical performance. The test involved capturing real-time data related to the number of triangles and corresponding frame rates within a dynamically changing environment. The analysis

confirmed that the model delivers a stable, high-quality visual experience without compromising system responsiveness.

### Justification of validation

The validation strategy employed is comprehensive, integrating both theoretical analysis and practical testing. By using a combination of benchmark comparisons, stress testing, and real-world applications, the model's reliability, performance, and adaptability were rigorously evaluated. The consistency in maintaining high frame rates and rendering quality across diverse scenarios supports the effectiveness of the proposed approach, making it well-suited for complex virtual environments. These validation results are crucial for confirming the model's potential for broader applications in fields such as entertainment, education, and virtual tours.

## CONCLUSION AND FUTURE WORK

Virtual reality technology has had a profound impact on various aspects of human life, significantly altering how people engage with digital environments. As society becomes more integrated with virtual spaces, the boundaries between physical and digital identities continue to blur. This growing reliance on virtual environments suggests a future where digital and virtual experiences are central to daily life.

The development of VR animation is expected to be diverse and full of potential, offering numerous opportunities for innovation. This article has contributed to this evolving field by combining Java and VRML to enhance the security and compatibility of 3D animation production processes. By applying fuzzy model recognition techniques, the frame rate of 3D animations has been improved, resulting in a more immersive and realistic experience. Additionally, VRML technology has been effectively employed in the creation of 3D models and the design of post-production roaming scenes, with careful attention to the precise handling of parameters and special effects.

The innovations presented in this study, particularly the application of fuzzy model recognition in conjunction with VRML, mark a significant step forward in the development of high-performance, interactive 3D animations. These contributions not only address current technical challenges but also lay the groundwork for more advanced virtual reality applications in entertainment, education, and beyond.

While the current application of these technologies in 3D animation is still in its formative stages, there is considerable room for growth. The work presented here represents an initial step, with many aspects still relatively simple, indicating clear pathways for future research and development. In an age characterized by rapid technological advancements, the potential for 3D animation to evolve is vast. By continuing to refine and apply these technologies, we can create increasingly sophisticated and visually compelling works.

However, to fully realize the potential of 3D animation, ongoing research and innovation are essential. This is the only way to achieve a seamless integration of 3D animation technology with artistic expression, paving the way for the development of even more advanced techniques. The article also underscores the importance of real-time animation

control, noting that the use of VRML has resulted in more accurate and natural simulation effects, which are particularly valuable in applications such as industrial process control.

Looking ahead, it will be important to further explore the final stages of animation synthesis, especially with the use of post-production tools, to unlock the full potential of these technologies. As the field of 3D animation continues to advance, the intersection of technology and art will be crucial in shaping new creative possibilities and pushing the boundaries of what can be achieved.

### Funding
The Shaanxi Provincial Art and Science Planning Project sponsored this research study. The project's name is Research on Modern Adaptability of Shaanxi Intangible Cultural Heritage Traditional Crafts. The project number is SY2021023. The funders had no role in study design, data collection and analysis, decision to publish, or preparation of the manuscript.

### Grant Disclosures
The following grant information was disclosed by the authors:
Shaanxi Provincial Art and Science Planning Project.
Research on Modern Adaptability of Shaanxi Intangible Cultural Heritage Traditional Crafts: SY2021023.

### Competing Interests
The authors declare there are no competing interests.

### Author Contributions
- Yu Zhu conceived and designed the experiments, performed the experiments, analyzed the data, performed the computation work, prepared figures and/or tables, authored or reviewed drafts of the article, and approved the final draft.
- Shifan Xie conceived and designed the experiments, performed the experiments, analyzed the data, performed the computation work, prepared figures and/or tables, authored or reviewed drafts of the article, and approved the final draft.

### Data Availability
The data and code are available in the Supplementary File.

### Supplemental Information
Supplemental information for this article can be found online at http://dx.doi.org/10.7717/peerj-cs.2354#supplemental-information.

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
