# Peer review of "Simulation methods realized by virtual reality modeling language for 3D animation considering fuzzy model recognition"

_PeerJ Computer Science, doi:10.7717/peerj-cs.2354_

## Round 0.1 · original submission · Major Revisions

The experts opinion has been received and you will see that some revisions need to be addressed. Therefore please update the paper in light of experts opinion and also:

* Improve the language of the paper
* Add justification about the novelty for the paper
* Add justification of validation for the proposed model.

Please carefully revise and resubmit.

Reviewer 1 ·

Basic reporting

Overall paper need proof reading and professional English writing suitable for journal. Introduction and conclusion need significant improvement to convince reader that a real & challenging problem is addressed by authors.

Abstract: Abstract can be updated with relevant academic vocabulary. such as first line "pays more and more" could be replaced with more appropriate terminologies.

Introduction:
1-References are missing. Authors must include relevant and updated references in this section to highlight significance of their work.
2- "Icing on the cake" on line 40 is must be replaced with academic research writing style and vocabulary.
3- objective of research not define din introduction section.
4- Authors need to clearly state the problem they are going to address and why is it significant.

Conclusion: Authors need to reemphasize the importance of research work. Avoid ambiguous sentences such as line 470 "However, the final animation synthesis depends on AE software, which the article should mention in depth." What authors means is not clear.

References: Authors need to rigorously review the existing literature and improve formatting.

Experimental design

The experimental design is pretty straightforward.

Validity of the findings

Impact and novelty is not fully addressed.

Additional comments

Paper is more suitable for a conference.

Reviewer 2 ·

Basic reporting

Authors should provide further in-depth descriptions of the technical implementations, particularly the VRML improvements and fuzzy model recognition. By elaborating on these elements, readers will better comprehend the innovations and their significance.

Experimental design

A more organized approach is required for the methodology section. The details of the VRML script integration and how fuzzy model recognition was used to enhance animation features are outlined here.

The data analysis concerning the connection between the frame rate and the number of triangles in virtual animation scenes could be more detailed. To clearly show the benefits of your approach, include graphs, statistical analysis, and comparisons to other techniques.
While it is laudable to maintain a frame rate of more than 40 frames per second, additional information on additional evaluation metrics could provide a more comprehensive understanding of the system's functioning. Latency, user experience ratings, and comparison with industry standards are factors to consider.

Validity of the findings

Provide a comparison analysis with current technologies and methods in 3D animation and virtual reality. Highlight the distinct features of your method and how it differs from or enhances current practices.
Through a summary of the significant findings, the significance of the contributions, and the wider implications for the field of 3D animation and VR, the conclusion can be strengthened.

Additional comments

Proofread the paper to improve readability in general and fix grammatical mistakes. As an example, the phrase "VRML can create 3D animation stereoscopic shapes and scenes and can map on the model, add lighting effects, establish user event response, and other better 3D animation stereo interaction" could be simplified to make it clearer.

---

## Round 0.2 · accepted · Accept

Dear authors

Thank you for resubmitting your paper after making the necessary changes. Based on the input from the experts I'm pleased to inform you that your manuscript is accepted.

Thank you

Reviewer 1 ·

Basic reporting

The authors have improved the document and had covered all important components.

Experimental design

The article is revised with sufficient information.

Validity of the findings

Findings arejustified.

Reviewer 2 ·

Basic reporting

The authors have substantially improved the manuscript in response to the reviewers' comments. My original concerns have been adequately addressed, and the revised version is now clear, concise, and well-supported. I am happy to recommend the manuscript for publication

Experimental design

The authors have substantially improved the manuscript in response to the reviewers' comments. My original concerns have been adequately addressed, and the revised version is now clear, concise, and well-supported. I am happy to recommend the manuscript for publication

Validity of the findings

The authors have substantially improved the manuscript in response to the reviewers' comments. My original concerns have been adequately addressed, and the revised version is now clear, concise, and well-supported. I am happy to recommend the manuscript for publication